# Discovery of Anti-MRSA Secondary Metabolites from a Marine-Derived Fungus *Aspergillus fumigatus*

**DOI:** 10.3390/md20050302

**Published:** 2022-04-28

**Authors:** Rui Zhang, Haifeng Wang, Baosong Chen, Huanqin Dai, Jingzu Sun, Junjie Han, Hongwei Liu

**Affiliations:** 1Key Laboratory of Structure-Based Drug Design & Discovery of Education, College of Traditional Chinese Materia Medica, Shenyang Pharmaceutical University, Shenyang 110016, China; zhangrui950714@163.com (R.Z.); wanghaifeng0310@163.com (H.W.); 2State Key Laboratory of Mycology, Institute of Microbiology, Chinese Academy of Sciences, Beijing 100101, China; chenbs@im.ac.cn (B.C.); daihq@im.ac.cn (H.D.); sunjz@im.ac.cn (J.S.)

**Keywords:** methicillin-resistant *Staphylococcus aureus*, *Aspergillus fumigatus*, chemical diversity, chemical ecology

## Abstract

Methicillin-resistant *Staphylococcus aureus* (MRSA), a WHO high-priority pathogen that can cause great harm to living beings, is a primary cause of death from antibiotic-resistant infections. In the present study, six new compounds, including fumindoline A–C (**1**–**3**), 12*β*, 13*β*-hydroxy-asperfumigatin (**4**), 2-*epi*-tryptoquivaline F (**17**) and penibenzophenone E (**37**), and thirty-nine known ones were isolated from the marine-derived fungus *Aspergillus fumigatus* H22. The structures and the absolute configurations of the new compounds were unambiguously assigned by spectroscopic data, mass spectrometry (MS), electronic circular dichroism (ECD) spectroscopic analyses, quantum NMR and ECD calculations, and chemical derivatizations. Bioactivity screening indicated that nearly half of the compounds exhibit antibacterial activity, especially compounds **8** and **1****1**, and **33**–**38** showed excellent antimicrobial activities against MRSA, with minimum inhibitory concentration (MIC) values ranging from 1.25 to 2.5 μM. In addition, compound **8** showed moderate inhibitory activity against *Mycobacterium bovis* (MIC: 25 μM), compound **10** showed moderate inhibitory activity against *Candida albicans* (MIC: 50 μM), and compound **13** showed strong inhibitory activity against the hatching of a *Caenorhabditis elegans* egg (IC_50_: 2.5 μM).

## 1. Introduction

Methicillin-resistant *Staphylococcus aureus* (MRSA) is recognized as one of the most common bacteria in both community and hospital-acquired infections, causing significant morbidity and mortality [1]. Compared to non-resistant *Staphylococcus aureus* infections, the mortality rate of MRSA infections increases by 64% [2]. Vancomycin is a last-resort treatment for MRSA infections. However, strains that are less susceptible to vancomycin are emerging in clinics [3,4]. As a result, new antibiotics to treat MRSA infections are desperately needed. In 2017, the development of new antibiotics for the treatment of MRSA infections is listed as a high urgency level by the WHO (World Health Organization) [5].

The marine environment is one of the most complex atmospheres on the earth, due to the huge variations in predation, temperature, pressure, light, and nutrient circumstances, etc. [6]. The organisms that thrive in marine environments could produce extremely diverse and complicated functional secondary metabolites that differ from those observed in terrestrial environments [6,7,8]. In recent decades, an increasing number of bioactive marine natural products (MNPs) have piqued the interest of chemists and pharmacologists for their medicinal values [9,10], such as the earliest marine sponge-derived anticancer drug cytarabine (Cytosar-U^®^), the marine sponge-derived antiviral drug vidarabine (Arasena A^®^), the mollusk-derived ziconotide (Prialt^®^) for the treatment of neuropathic pain, the famous sponge-derived anticancer drugs trabectedin (Yondelis^®^) and eribulin mesylate (Halaven^®^), and the marine cyanobacterium-derived anticancer drug disitamab vedotin (Aidixi™) and tisotumab vedotintftv (TIVDAK™), etc. [11,12,13,14,15,16,17].

Marine fungi have been shown to produce a variety of secondary metabolites with a variety of structures and bioactivities [18], including antibacterial, antiviral, anticancer, and anti-inflammatory characteristics, and have already provided a number of promising leads against MRSA [19,20]. Pestalone is a well-known anti-MRSA compound that was discovered by Fenical and colleagues, after co-culturing a fungus of the genus *Pestalotia* with a unicellular marine bacteria (strain CNJ-328) [21,22].

In our search for new anti-MRSA agents from marine-derived fungi, the EtOAc extract of the fungus *Aspergillus fumigatus* H22 was found to show strong anti-MRSA activity by in vitro anti-MRSA assay. A chemical investigation on its extract led to the identification of 45 secondary metabolites (Figure 1), including six new novel compounds, including fumindoline A–C (**1**–**3**), 12*β*,13*β*-hydroxy-asperfumigatin (**4**), 2-*epi*-tryptoquivaline F (**17**), and penibenzophenone E (**37**). The isolation and structure characterization of the new compounds, as well as the antibacterial activity of all the compounds, are described in this work.

## 2. Results

### Structure Elucidation of the Isolated Compounds

Fumindoline A (**1**) was obtained as a chartreuse powder and had the molecular formula of C_21_H_23_N_3_O_4_, based on HRESIMS data (Appendix A), corresponding to 12 indices of hydrogen deficiency. This molecular formula was corroborated by ^1^H NMR and ^13^C NMR spectroscopic data. The ^1^H NMR data (Table 1) showed characteristic signals for a 1,2,4-trisubstituted benzene ring (*δ* 8.21 (d, *J* = 8.7 Hz), 6.91 (dd, *J* = 8.7 and 2.3 Hz), and 7.05 (d, *J* = 2.3 Hz)), two singlet olefinic protons (*δ*_H_ 8.59 (s) and 6.79 (s)), three singlet methyl groups (*δ*_H_ 2.08 (s), 2.17 (s), and 3.89 (s)), and two exchangeable (*δ* 8.42 (br. s) and 11.85 (br. s)). The ^13^C NMR and HSQC data of **1** revealed the presence of twenty-one carbon resonances, including three methyls (*δ*_C_ 20.6, 27.3, and 55.4), three methylenes (*δ*_C_ 24.9, 31.2, and 38.3), five *sp*^2^ methines (*δ*_C_ 94.8, 110.2, 111.1, 118.5, and 123.1), and ten nonprotonated carbons (eight *sp*^2^ carbons at *δ*_C_ 115.0, 128.9, 135.1, 138.7, 142.4, 142.8, 158.3, and 160.7; one amide carbonyl carbon at *δ*_C_ 164.7, and one carboxyl carbon at *δ*_C_ 174.2).

The planar structure of **1** was defined by the 2D NMR spectra, particularly the ^1^H–^1^H COSY and HMBC data (Appendix A). The HMBC correlations from H-9 to C-8, C-11, C-12, and C-13, from H-10 to C-8, and C-11, and from H-12 to C-8, C-10, C-11, and C-13, together with the ^1^H–^1^H COSY correlations of H-9/H-10/H-12, which indicated a 1,2,4-trisubstituted benzene. The ^1^H–^1^H COSY correlations of *N*H-15/H_2_-16/H_2_-17/H_2_-18, as well as the HMBC correlations from H_2_-16 to C-14, C17, and C18, from H_2_-17 to C-16, C-18, and C-19, from H_2_-18 to C-16, C-17, and C-19, led to the identification of the *γ*-aminobutyric acid residue. The HMBC correlations from H-6 to C-2, C-7, C-8, and C-14, H-9 to C-7, and H_2_-16 to C-14, as well as the chemical shifts of C-2 (*δ* 135.1), C-3 (*δ* 142.4), C-5 (*δ* 158.3), C-6 (*δ* 111.1), and C-14 (*δ* 164.7), supported a 2-pyridinecarboxylic acid moiety that was connected with a *γ*-aminobutyric acid moiety through C-14 and linked with a 1,2,4-trisubstituted benzene moiety through C-7, and completed the assignment of the moiety. The HMBC correlations from H-20 to C-2, C-3, and C-21, together with the ^1^H–^1^H COSY correlations of H-20/H_3_-22/H_3_-23, suggested that the isobutenyl group was located at C-3 of the 2-pyridinecarboxylic acid moiety. The key HMBC correlations from H_3_-24 to C-11 indicated that the methoxy group was located at C-11. Furthermore, these data accounted for 11 of the 12 degrees of unsaturation, implying the presence of an additional cycle, attributed to the *N*H bridging between C-2 and C-13 to establish the indole-pyridinecarboxylic acid skeleton (Figure 2). Therefore, the 2D structure of **1** was determined as shown below.

Fumindoline B (**2**) was obtained as a chartreuse powder. Its molecular formula, C_22_H_23_N_3_O_4_, with 14 degrees of unsaturation, was established on the basis of the HRESIMS data (Appendix A). The UV spectrum showed absorptions at 282 nm and 343 nm, which were similar to those of **1**, indicating that **2** might have the same conjugation system as **1**. The IR spectrum indicated the presence of a secondary amine *N*-H signal (2980 cm^−1^) and an amide carbonyl signal (1628 cm^−1^). The ^1^H NMR and ^13^C NMR spectra indicated the presence of two sets of very similar signals, with the same number of carbons (Appendix A). The spectra of the two sets of signals are well resolved in pairs at 313K and 298K in DMSO-*d*_6_, indicating the presence of two relatively stable isomers. From the integrals of the completely resolved signals, a ratio of 1:0.7 was calculated for the two stable isomers. To be better distinguished, we assigned the major isomer as **2a** and the minor one as **2b**, respectively (Figure 3).

The ^1^H NMR and ^13^C NMR spectra data of **2** showed close similarity to those of **1**, with the biggest difference in the methine (CH-19). A detailed analysis of the 2D NMR data, including HSQC, HMBC, and ^1^H–^1^H COSY spectra, revealed that **2** contained the same indole-pyridinecarboxylic acid skeleton as that of **1** (Figure 4 and Appendix A). The HMBC correlations from H_2_-16 to C-17 and C-18, H_2_-17 to C-16, C-18, and C-19, H_2_-18 to C-17, C-19, and C-20, and the ^1^H–^1^H COSY correlations of H_2_-16/H_2_-17/H_2_-18/H-19, together with the molecular formula, indicated the presence of a proline moiety, and this conclusion was also confirmed by the 14 degrees of unsaturation and the chemical shifts of C-16 (*δ*_C_ 49.6 (**2a**); *δ*_C_ 47.5 (**2b**)) and C-19 (*δ*_C_ 59.8 (**2a**); *δ*_C_ 60.4 (**2b**)).

The *E*/*Z* isomer exists in the tertiary amide. In the solution at room temperature, the slow rotation of the C–*N* bond in NMR makes it possess the characteristics of a partial double bond [23]. A comparison of the ^1^H NMR signals and ^13^C NMR signals of **2a** and **2b** revealed differences in the proline moiety, including variations in the H-19 (*δ*_H_ 4.48 (**2a**); *δ*_H_ 5.3 (**2b**)), C-16 (*δ*_C_ 49.6 (**2a**); *δ*_C_ 47.5 (**2b**)), C-17 (*δ*_C_ 25.2 (**2a**); *δ*_C_ 21.9 (**2b**)), and C-18 (*δ*_C_ 28.6 (**2a**); *δ*_C_ 31.2 (**2b**)). As shown in Figure 4, strong NOE effects between H-6 and H-16 for **2a** and between H-6 and H-19 for **2b** were observed in the ROSEY spectrum (Appendix A). 

The absolute configuration of the amino acids from compound **2** was determined by the advanced Marfey’s method [24]. The mixture obtained after hydrolyzing compound **2** and further derivatization with l-FDAA was analyzed by HPLC-DAD. The HPLC analyses of the mixture of hydrolysates and appropriate amino acid standards confirmed the l configurations for proline in **2** (Figure 5). Consequently, the absolute configuration of **2** was elucidated to be 19*S*.

Fumindoline C (**3**) was obtained as a chartreuse powder. The molecular formula of **3** was established to be C_23_H_25_N_3_O_4_ from its HREIMS data (Appendix A). The ^1^H and ^13^C NMR spectra of **3** were similar to those of **2**, possessing two sets of signals (Appendix A), except for the presence of an additional methoxyl group. The substitution of the methoxyl group was further confirmed by the HMBC correlations from H_3_-26 to C-20. A further comprehensive analysis of its ^1^H–^1^H COSY, HSQC, and HMBC spectra assigned the planar structure of **3** (Appendix A). The relative configuration of **3** was determined to be the same as that of **2** by their similar structure and ROESY data (Appendix A). Accordingly, **3** was determined to be a methyl ester of **2**.

12*β*,13*β*-hydroxy-asperfumigatin (**4**) was obtained as a white amorphous solid. Its molecular formula was determined as C_27_H_33_N_3_O_7_ by HRESIMS data (Appendix A). The ^1^H NMR spectrum of **4** (Appendix A) displayed four singlet methyl groups (*δ*_H_ 1.17, 1.25, 2.11, and 2.21), one methoxyl group (*δ*_H_ 3.85) and four olefinic/aromatic protons (*δ*_H_ 6.40, 6.90, 7.27, and 7.45). The ^13^C-NMR spectrum (Appendix A) exhibited 27 carbon resonances accounted for the functional groups described above and three amide carbonyl carbons (*δ*_C_ 164.7, 165.5, and 165.9). A comprehensive analysis of its 2D NMR spectra, including ^1^H-^1^H COSY, HSQC, and HMBC experiments, confirmed the planar structure of **4** (Table 2, Appendix A), revealing the presence of the indole moiety and the diketopiperazine moiety in **4** (Figure 6). The planar structure of **4** was determined to be the same as that of asperfumigatin (**5**), by detailed interpretation of the 2D NMR spectra and NMR data comparison between **4** and **5**. Considering the same biosynthesis origin, compound **4** was deduced to share the same absolute configuration at C-3 and C-6 as those of **5**–**13**.

Owing to a lack of sufficient ROESY correlations, the relative configurations of C-12 and C-13 were not determined (Appendix A). The relative configurations of **4** were determined by the DP4+ probability, based on a theoretical NMR calculation that has been proven to be a very powerful tool in natural product structure elucidation [25,26]. The NMR shifts of eight possible relative orientation isomers were calculated with the GIAO method at the MPW1PW91/6-31+G(d,p), and the DP4+ probabilities of each configuration were evaluated based on Boltzmann-averaged theoretical NMR shielding tensors, which provided a 91.55% confidence for the relative configuration 3*S**, 6*S**, 12*S**, 13*R** (Appendix A).

To determine the absolute configurations of **4**, a ECD calculation method was applied. The two configurations (3*S*, 6*S*, 12*S*, 13*R*)-**4** and (3*R*, 6*R*, 12*R*, 13*S*)-**4** were calculated using time-dependent density functional theory (TDDFT) at PBE1PBE/6-311 G* level, with the PCM model in methanol, and corrected with a 2 nm blue shift according to UV data. A comparison of the experimental ECD spectrum of **4** and the calculated ECD spectra of (3*S*, 6*S*, 12*S*, 13*R*)-**4** and (3*R*, 6*R*, 12*R*, 13*S*)-**4** showed that the experimental ECD spectrum of **4** is consistent with the calculated ECD spectrum of (3*S*, 6*S*, 12*S*, and 13*R*)-**4** (Figure 7). Thus, the absolute configuration of **4** was assigned as 3*S*, 6*S*, 12*S*, and 13*R*, and named as 12*β*,13*β*-hydroxy-asperfumigatin. The only difference between compound **4** and compound **5** is the orientation of the two hydroxyl groups (12-OH, 13-OH).

2-*epi*-tryptoquivaline F (**17**), which was isolated as a white amorphous solid, exhibited the [M + H]^+^ peak at *m*/*z* 403.1399 (HRESIMS), corresponding to C_22_H_18_N_4_O_4_, as well as sixteen degrees of unsaturations (Appendix A). The ^1^H NMR, ^13^C NMR and HSQC spectra (Table 3, Appendix A) of **17** revealed the presence of two 1, 2-disubstituted benzene rings (*δ*_H_*/δ*_C_ 7.31 (d, *J* = 7.5 Hz)/124.4, 7.22 (dd, *J* = 8.0 and 7.5 Hz)/125.8, 7.43 (dd, *J* = 8.0 and 7.5 Hz)/131.3, 7.65 (d, *J* = 8.0 Hz)/115.8; *δ*_H_*/δ*_C_ 8.29 (d, *J* = 8.1 Hz)/126.8, 7.58 (dd, *J* = 8.1 and 7.5 Hz)/128.2, 7.85 (dd, *J* = 8.1 and 7.5 Hz)/135.4, 7.78 (d, *J* = 8.1 Hz)/128.0), one methyl (*δ*_H_*/δ*_C_ 1.28 (3H, d, *J* = 6.5 Hz)/17.9), one methylene (*δ*_H_ 2.61 (dd, *J* = 13.4 and 10.5 Hz), 3.68 (dd, *J* = 13.4 and 4.3 Hz), *δ*_C_ 33.4), four nitrogenated methines (*δ*_H_*/δ*_C_ 4.26 (d, *J* = 6.5 Hz)/58.5, 5.03 (dd, *J* = 4.3 and 10.5 Hz)/58.3, 5.82 (s)/82.6, 8.11 (s)/145.6), eight quaternary carbons including three carbonyls (*δ*_C_ 172.4, 170.9, and 161.0), four aromatic or olefinic carbon atoms (*δ*_C_ 121.9, 134.4, 138.9, and 148.1), and one oxygenated one (*δ*_C_ 91.3). The NMR data of compound **17** were similar to those of tryptoquivaline F [27], indicating the presence of one 6-5-5 gem-methyl imidazoindolone ring and one quinazoline-4-one moiety.

The partial relative configuration of **17** was confirmed by a ROESY experiment (Appendix A). The ROESY correlations of H-2 (*δ*_H_ 5.82, s) with H-15 (*δ*_H_ 4.26, q, *J* = 6.5 Hz) indicated that H-2 and H-15 were on the same face, while H_3_-27 (*δ*_H_ 1.28, d, *J* = 6.5 Hz) were on the opposite face (Figure 8). The relative configurations of C-2 and C-15 were assigned as 2*S* and 15*S*. However, owing to a lack of sufficient ROESY correlations, neither the orientation of C-3 nor C-12 could be determined.

Similar to compound **4**, the NMR shifts of four relative configuration isomers (2*S*, 3*S*, 12*R*, 15*S*; 2*S*, 3*R*, 12*S*, 15*S*; 2*S*, 3*R*, 12*R*, 15*S*; 2*S*, 3*S*, 12*S*, 15*S*) were calculated and the DP4+ probability, based on a theoretical NMR calculation, was applied. The 100% DP4+ probability for **17a** revealed that the relative configuration of **17** was 2*S**, 3*S**, 12*R** and 15*S** (Appendix A).

The absolute configurations of **17** were deduced by the comparison of the experimental and simulated ECD spectra generated by TDDFT at B3LYP/6-311+G(2d,p) level with the PCM model in methanol and corrected -5 nm according to the UV data. A comparison of the observed ECD spectra for **17,** with the calculated ECD spectra for the (2*S*, 3*S*, 12*R*, 15*S*)-**17** and (2*R*, 3*R*, 12*S*, 15*R*)-**17** enantiomers, is shown in Figure 9. The overall ECD spectra for (2*S*, 3*S*, 12*R*, 15*S*)-**17** are in good accordance with the experimental ECD for **17**. Thus, compound **17** was determined to be 2*S*, 3*S*, 12*R*, and 15*S*. The differences between **17** and tryptoquivaline F are the configuration of C-2. Therefore, compound **17** was identified as 2-*epi*-tryptoquivaline F.

Compound **37** was isolated as a yellowish powder. Its molecular formula was determined as C_17_H_16_O_7_ based on the HRESIMS (Appendix A), implying ten degrees of unsaturation. The ^1^H NMR spectrum (Appendix A) of **37** showed one hydrogen-bonded phenol moiety at *δ*_H_ 13.55 (s, 6′-OH), four aromatic methine protons at *δ*_H_ 7.19 (br. s, H-5), 6.88 (br. s, H-3), 5.90 (d, *J* = 2.2 Hz, H-5′) and 5.80 (d, *J* = 2.2 Hz, H-3′) for two sets of AB meta-coupling, two methoxy groups at *δ*_H_ 3.64 (s, 8-OMe) and 3.26 (s, 2′-OMe), and a methyl group at *δ*_H_ 2.30 (s, 4-Me). A comparison of its ^1^H NMR and ^13^C NMR spectra (Table 4) with those of sulochrin (**38**) suggested the same benzophenone skeleton between them [28]. The HMBC correlations from the proton of 6′-OH to C-6′, C-1′ and C-5′ indicate that 6′-OH was located at C-6′. The HMBC correlations from 2′-OCH_3_ to C-2′, 4-CH_3_ to C-3, C-4, C-5, and 8-CH_3_ to C-7 indicate that the two methoxy groups and one methyl group were located at C-2′, C-7 and C-4, respectively. In addition, the HMBC correlations from H-3 to C-1, C-2, C-5, from H-3′ to C-1′, C-2′, C-4′, C-5′, from H-5 to C-1, C-3, C-6 and C-7, and from H-5′ to C-1′, C-3′ and C-6′ confirmed the proposed structure (Figure 10). Therefore, compound **37** was determined as penibenzophenone E.

Other known compounds were identified as asperfumigatin (**5**) [29], demethoxyfumitremorgin C (**6**) [30], fumitremorgin C (**7**) [30], 12,13-dihydroxyfumitremorgin C (**8**) [31], 12*α*-hydroxy-13-oxofumitremorgin C (**9**) [32], fumitremorgin B (**10**) [33], 13-oxofumitremorgin B (**11**) [34], cyclotryprostatin B (**12**) [35], verruculogen (**13**) [36], 6-methoxyspirotryprostatin B (**14**) [37], (−)-spirotryprostatin A (**15**) [38], spirotryprostatin C (**16**) [39], fumiquinazoline C (**18**) [40], (+)-alantrypinone (**19**) [41,42], oxoglyantrypine (**20**) [43], (−)-chaetominine (**21**) [44], 11-epi-chaetominine (**22**) [29], fumigaclavine C (**23**) [45,46], bisdethiobis(methylthio)gliotoxin (**24**) [47], pyripyropene A (**25**) [48], pseurotin F1 (**26**) [49], pseurotin F2 (**27**) [49], pseurotin A (**28**) [50], 11-O-methylpseurotin A (**29**) [51], azaspirofuran B (**30**) [52], azaspirofuran A (**31**) [52], fumagiringillin (**32**) [53], fumagillin (**33**) [54], helvolic acid (**34**) [55], 6-O-propionyl-16-O-deacetylhelvolic acid (**35**) [55], 16-O-propionyl-6-O-deacetylhelvolic acid (**36**) [55], sulochrin (**38**) [28], monomethylsulochrin (**39**) [56], 8′-O-methylasterric acid (**40**) [29], dimethyl 2,3′-dimethylosoate (**41**) [56], questin (**42**) [57], (+)-2′***S***-isorhodoptilometrin (**43**) [58], 6-hydroxy-8-methoxy-3-methylisocoumarin (**44**) [59], and trypacidin (**45**) [60], based on the spectroscopic analyses and in comparison with the literature data.

The antibacterial activities of the isolated compounds were determined against methicillin-resistant *Staphylococcus aureus* (MRSA) (clinical isolate strain), vancomycin-resistant enterococci *E. faecalis* (VRE), *Candida albicans* SC5314, *Mycobacterium bovis* ATCC35743 constitutive GFP expression (pUV3583c-GFP), and *Escherichia coli* O57:H7, within 100 μM. The results showed that nearly half of the compounds exhibit antibacterial activity (Table 5), especially compounds **5**, **8**, **10**, **11**, **16**, **21**, **23**, **29**–**38**, and **41** exhibited antimicrobial activities against MRSA, with minimum inhibitory concentration (MIC) values ranging from 1.25 to 25 μM. Furthermore, compound **8** also exhibited strong activity against *M. bovis* with a MIC of 25 μM, compound **10** showed moderate activity against *C. albicans* with a MIC of 50 μM. Moreover, compound **13** inhibited the egg hatching of *Caenorhabditis elegans* with a IC_50_ of 2.5 μM.

## 3. Discussion

The marine environmental stress conditions induce many faunae and symbiont microorganisms to synthesize and release secondary metabolites of unique structures and interesting biological activities [61]. These bioactive compounds can serve as an important source for drug discovery. Marine-derived fungi are important sources for the discovery of new antibacterial natural products. Wang et al. isolated the *Chaetomium* sp. strain NA-S01-R1 from a deep-sea (4050 m) fungus that produced novel chlorinated azaphilone polyketides with antibacterial activity against MRSA [62]. The *Emericellopsis minima* strain A11, isolated from Talcahuano Bay (Chile), produced an antibiotic called emerimicin IV, with moderate activity against clinical isolates of MDR vancomycin-resistant strains of *E. faecalis* and MRSA with MIC of 12.5 μg/mL and 100 μg/mL, respectively [63].

*A. fumigatus* belongs to the filamentous fungi family that is widely distributed in all environments and can cause many diseases and life-threatening conditions in immunocompromised patients [64]. *A. fumigatus* can produce a wide array of secondary metabolites due to its remarkable adaptability to different environmental conditions, such as fumitremorgins, fumagillins, pseurotins, fumigaclavines, gliotoxins, and helvolic acid derivatives.

Inspired by chemical ecology, we found a marine fungus *A. fumigatus* H22 with strong antibacterial activities from the marine fungi library. Through in-depth chemical mining, we found 45 compounds, including 6 new compounds, from the culture of this fungus. A evaluation of biological activity showed that nearly half of the compounds exhibit antimicrobial activity. Fumitremorgins derivatives (**4**-**11**) have very similar structures, but only a few have strong anti-MRSA activity. Compounds **5**, **8** and **11** with strong anti-MRSA activity contain hydroxyl group at C-13, while compounds **6** and **7** without anti-MRSA activity have no hydroxyl group at C-13. In addition, compounds **4** and **5** have the same planar structure, but the 13-OH of compound **4** without anti MRSA activity was α-oriented, while compound **5** and other strongly active compounds were *β*-oriented. Therefore, it is preliminarily speculated that there is a certain correlation between the substituents and stereoconfiguration in C-12 and C-13 and their anti MRSA activity. Fumitremorgin B (**10**) was reported with antifungal activity against a variety of phytopathogenic fungi, but it showed weak activity against vancomycin-resistant *E. faecalis* (VRE), *M. bovis*, and *E. coli* in our in vitro assay, which could be involved in fighting against invasion by other pathogens [65]. 

Pseurotins, with a unique heterospirocyclic furanone-lactam structure, exhibit a broad range of biological activities. In addition to antifungal and antibiotic activities [66,67], pseurotins were also shown to regulate enzymes of cellular metabolism [68], to possess anti-angiogenic activity, to modulate cell differentiation [69], and to inhibit endothelial cell migration [70,71,72]. Fumagillin (**33**) have been demonstrated to have antitumor, antibacterial and antiparasitic effects [73]. Previous studies revealed that helvolic acid (**34**) exhibited in vitro antimalarial activity against multidrug resistant *Plasmodium falciparum* [74], antitrypanosomal activity against *Trypanosoma brucei* [75], and antimycobacterial activity against *M. tuberculosis* H37Ra [76]. Our current research showed the strong activities of oxofumitremorgin B (**11**), helvolic acid (**34**), 6-O-propionyl-16-O-deacetylhelvolic acid (**35**), 16-O-propionyl-6-O-deacetylhelvolic acid (**36**), sulochrin (**38**) and 8′-O-methylasterric acid (**40**) against MRSA, with a MIC of 1.25 μM. 

From our current findings, it can be found that *A. fumigatus* from marine sources can produce rich bioactive secondary metabolites, especially in anti-MRSA.

## 4. Materials and Methods

### 4.1. General

UV data, optical rotation, and IR data were recorded on Genesys-10S UV–Vis spectrophotometer (Thermo Fisher Scientific, Waltham, MA, USA), MCP 200 automatic polarimeter (Anton Paar, Graz, Austria), and IS5 FT-IR spectrophotometer (Thermo Fisher Scientific, Waltham, MA, USA), respectively. NMR spectral data were obtained with a Bruker AVANCE-500 spectrometer (Bruker, Bremen, Germany) (DMSO-*d*_6_, *δ*_H_ 2.50/*δ*_C_ 39.52, and CDCl_3_, *δ*_H_ 7.26/*δ*_C_ 77.16). High-resolution electrospray ionization mass spectrometry (HRESIMS) data were obtained on an Agilent Accurate-Mass-Q-TOF LC/MS 6520 instrument (Agilent Technologies, Santa Clara, CA, USA). The CD spectra were measured by JASCO J-815 spectropolarimeter (JASCO, Tsukuba, Japan). Silica gel (Qingdao Haiyang Chemical Co., Ltd., Qingdao, China, 200–300 mesh), ODS (octadecylsilyl, 50 μM, YMC Co., Ltd., Kyoto, Japan), and Sephadex LH-20 (GE Healthcare, Uppsala, Sweden) were used for column chromatography. Semi-preparative HPLC was performed on an Agilent 1200 HPLC system equipped with an Agilent DAD UV–vis spectrometric detector (Agilent Technologies Inc., CA, USA), using a reversed-phase Eclipse XDB-C18 column (5 μM, 9.4 × 250 mm; Agilent, MA, USA), with a flow rate of 2.0 mL/min. The biological reagents, chemicals and media were purchased from standard commercial sources, unless stated.

### 4.2. Fungal Material

The fungus H22 was isolated from middle seawater from the Western Pacific. The sample (1 mL) was diluted with sterile H_2_O, 100 μL of which was deposited on a PDA (200 g of potato, 20 g of glucose, 20 g of agar per liter of seawater collected in the Western Pacific) plate containing chloramphenicol (100 μg/mL) and streptomycin (100 μg/mL) as a bacterial inhibitor. A single colony was transferred onto another PDA plate and was identified according to its morphological characteristics and 18*S* rRNA gene sequences. The phylogenetic tree (Appendix A), constructed from the ITS gene sequence, indicated that H22 belonged to the genus of *Aspergillus,* with the highest similarity to *A. fumigatus* (99.86%, accession number NRRL 163 s). In consideration of the morphological features and phylogeny (Appendix A), this fungus was identified as *A. fumigatus*. A reference culture of *A. fumigatus* H22 maintained at −80 °C was deposited in our laboratory.

### 4.3. Fermentation and Extraction

The isolate was grown for 7 days at 28 °C, on slants of a PDA medium. The spores of the strain on the plate were collected using 0.01% sterile Tween 80 (BTL, Warsaw, Poland) and adjusted to 1 × 10^6^ CFU/mL to make inoculum. A large-scale fermentation was carried out in 50 × 500 mL Fernbach culture flasks, holding 100 g of rice in 110 mL of distilled water (each with 0.5 mL of spore suspension) and incubated for 4 weeks at 28 °C. With the help of ultrasonication, the fermented rice substrates were extracted with ethyl acetate (3 × 5 L), and the organic solvent was filtered and evaporated to dryness under a vacuum to obtain the crude extract (78.0 g).

### 4.4. Isolation and Characterization Data

The ethyl acetate (EtOAc) fraction was subjected to silica gel column chromatography (CC), eluted with dichloromethane/acetone (D/A, *v*/*v*, 100:0, 100:1, 50:1, 30:1, 25:1, 20:1, 10:1, 5:1) and dichloromethane/methanol (D/M, *v*/*v*, 5:1, 2:1, 0:100) to give 10 fractions (HS.1–HS.10). 

HS.3 (4.94 g eluted with D/A, *v*/*v*, 50:1) was purified by RP-HPLC, using 37% acetonitrile in acidic water (0.01% TFA) to obtain compounds **45** (9.0 mg, *t*_R_ = 37.5 min), **42** (103.0 mg, *t*_R_ = 44.5 min) and **39** (19.2 mg, *t*_R_ = 47.6 min). 

Fraction HS.4 (5.98 g from D/A, *v*/*v*, 30:1) was separated by ODS, using a gradient from 20% to 100% methanol in water to afford 12 subfractions (HS.4-1–HS.4-12). HS.4-3 (203.0 mg) was further purified using C8-RP-HPLC on a Agilent Eclipse XDB-C8 (5 μM, 250 × 9.4 mm), with a gradient elution from 30% to 40% acetonitrile in 60 min to give compounds **21** (102.1 mg, *t*_R_ = 22.5 min), **18** (12.0 mg, *t*_R_ = 30.6 min), **17** (7.0 mg, *t_R_* = 39.6 min) and **41** (2.0 mg, *t*_R_ = 50.2 min). HS.4-4 (77 mg) was further purified using C8-RP-HPLC with 35% acetonitrile to give compounds **8** (3.5 mg, *t*_R_ = 33.6 min), **7** (3.5 mg, *t*_R_ = 41.5 min) and **31** (3.0 mg, *t*_R_ = 66.2 min). Compounds **13** (79.0 mg, *t*_R_ = 17.4 min), **10** (7.0 mg, *t*_R_ = 23.6 min) and **33** (5.0 mg, *t*_R_ = 25.8 min) were obtained from HS.4-9 (252 mg) by RP-HPLC, using 55% acetonitrile in acidic water.

Fraction HS.5 (5.43 g, from D/A, *v*/*v*, 25:1) was first separated by ODS, using a gradient from 30% to 100% methanol in water to afford HS.5-1–HS.5-11. Subfraction HS.5-4 (30 mg) was purified using RP-HPLC on a Agilent Eclipse XDB-C8 column (5 μM, 250 × 9.4 mm) with 40% acetonitrile in 20 min to give compounds **23** (5.0 mg, *t*_R_ = 4.1 min) and **39** (4.2 mg, *t*_R_ = 7.5 min). Compound **24** (5.0 mg, *t*_R_ = 24.8 min) was obtained from subfraction HS.5-4-5 (32 mg) by RP-HPLC, using 28% acetonitrile in acidic water (0.01% TFA). Compounds **22** (2.2 mg, *t*_R_ = 28.1 min) and **21** (3.0 mg, *t*_R_ = 29.7 min) were obtained from HS.5-4-8 (144 mg) by RP-HPLC, using 29% acetonitrile in acidic water (0.01% TFA).

Fraction HS.6 (6.72 g, from D/A, *v*/*v*, 20:1) was first separated by ODS, using a gradient from 20% to 100% methanol in water to afford HS.6-1–HS.6-17. HS.6-2 (289 mg) was purified using C8-RP-HPLC eluting with 50% to provide compound **44** (2.0 mg, *t*_R_ = 16.3 min). Compounds **37** (3.0 mg, *t*_R_ = 19.1 min) and **38** (1.5 mg, *t*_R_ = 20.2 min) were obtained from HS.6-4 (326.0 mg) by RP-HPLC, using 45% acetonitrile in acidic water. HS.6-5 (522 mg) was purified using RP-HPLC eluting with 50% acetonitrile to give compounds **15** (9.0 mg, *t*_R_ = 10.2 min), **9** (15.0 mg, *t*_R_ = 13.2 min), and **12** (100.0 mg, *t*_R_ = 14.6 min). Compound **16** (3.0 mg, *t*_R_ = 25.2 min) was obtained from subfraction HS.6-9 (17.5mg) by RP-HPLC, using 60% acetonitrile in acidic water. Compounds **33** (19.8 mg, *t_R_* = 10.7 min), **34** (2.0 mg, *t*_R_ = 12.3 min) and **35** (2.0 mg, *t*_R_ = 13.2 min) were obtained from HS.6-17 (365 mg) by RP-HPLC, using 70% acetonitrile in acidic water.

Fraction HS.7 (10.63 g, D/A, *v*/*v*, 10:1) was first separated by ODS, using a gradient from 35% to 100% methanol in water to afford HS.7-1–HS.7-13. Compounds **40** (6.0 mg) and **29** (2.0 mg) were obtained from HS.7-2 and HS.7-3 by recrystallization in methanol, respectively. Compounds **30** (2.0 mg, *t*_R_ = 9.1 min), **20** (2.0 mg, *t*_R_ = 11.1 min) and **6** (2.1 mg, *t*_R_ = 11.9 min) were obtained from HS.7-4 (11.2 mg) by RP-HPLC, using 65% acetonitrile in acidic water. Compounds **4** (3.2 mg, *t*_R_ = 14.9 min) and **5** (2.8 mg, *t*_R_ = 16.1 min) were purified from HS.7-7, using RP-HPLC with 50% acetonitrile. HS.7-9 (420.0 mg) was purified using C8-RP-HPLC with 65% methanol to give compounds **43** (2.0 mg, *t*_R_ = 12.5 min), **32** (20.2 mg, *t*_R_ = 16.2 min) and **19** (37.3 mg, *t*_R_ = 18.1 min). 

Fraction HS.8 (8.82 g, D/A, *v*/*v*, 5:1) was first separated by ODS, using a gradient from 20% to 100% methanol in water to afford 22 subfractions (HS.8-1–HS.8-22). HS.8-3 (100.0 mg) was further purified on C8-RP-HPLC eluting with 35% acetonitrile in acidic water to give compounds **28** (11.8 mg, *t*_R_ = 19.3 min), **27** (3.1 mg, *t*_R_ = 22.4 min), and **26** (2.8 mg, *t*_R_ = 24.3 min). HS.8-18 (124.0 mg) was purified on a C8-RP-HPLC eluting with a gradient elution from 70% methanol to give compounds **3** (15.0 mg, *t*_R_ = 12.5 min), **2** (5.0 mg, *t*_R_ = 15.2 min), **11** (20.0 mg, *t*_R_ = 18.4 min), **1** (5.0 mg, *t*_R_ = 19.2 min) and **25** (8.0 mg, *t*_R_ = 22.5 min).

Fumindoline A (**1**). UV (MeOH) *λ*_max_ (log *ε*) 286 (1.62), 345 (0.48). ^1^H NMR and ^13^C NMR, see Table 1, 2D NMR spectra, see Appendix A. Positive HRESIMS: *m/z* 382.1768 [M + H]^+^ (calcd for C_2__1_H_24_N_3_O_4_, 382.1761, Appendix A). 

Fumindoline B (**2**). Chartreuse powder; (α${)}_{D}^{25}$ −34.99 (*c* 0.1, MeOH); UV (MeOH) *λ*_max_ (log *ε*) 282 (2.82), 343 (0.82); ^1^H NMR and ^13^C NMR, see Table 1, 2D NMR spectra, see Appendix A; Positive HRESIMS: *m/z* 394.1765 [M + H]^+^ (calcd for C_2__2_H_24_N_3_O_4_, 394.1761, Appendix A).

Fumindoline C (**3**). Chartreuse powder; (α${)}_{D}^{25}$ −21.00 (*c* 0.1, MeOH); UV (MeOH) *λ*_max_ (log *ε*) 286 (1.62), 345 (0.50); ^1^H NMR and ^13^C NMR, see Table 1, 2D NMR spectra, see Appendix A; Positive HRESIMS: *m/z* 408.1916 [M + H]^+^ (calcd for C_23_H_26_N_3_O_4_, 408.1918, Appendix A).

12*β*,13*β*-hydroxy-asperfumigatin (**4**). White amorphous solid; (α${)}_{D}^{25}$ +26.00 (*c* 0.1, MeOH); UV (MeOH) *λ*_max_ (log *ε*) 222 (1.51), 270 (0.60); ^1^H NMR and ^13^C NMR, see Table 2, 2D NMR spectra, see Appendix A; Positive HRESIMS: *m/z* 494.2720 [M + H − H_2_O]^+^ (calcd for C_2__7_H_3__2_N_3_O_6_, 494.2726, Appendix A). 

2-*epi*-tryptoquivaline F (**17**). White amorphous solid; (α${)}_{D}^{25}$ +221.96 (*c* 0.1, CH_2_Cl_2_); UV (CH_2_Cl_2_) *λ*_max_ (log *ε*) 212 (2.21), 233 (1.69); ^1^H NMR and ^13^C NMR, see Table 3, 2D NMR spectra, see Appendix A; Positive HRESIMS: *m/z* 403.1399 [M + H]^+^ (calcd for C_2__2_H_1__9_N_4_O_4_, 403.1401, Appendix A). 

Penibenzophenone E (**37**). Yellowish powder; UV (MeOH) *λ*_max_ (log *ε*) 205 (3.28), 303 (1.63); ^1^H NMR and ^13^C NMR, see Table 4, 2D NMR spectra, see Appendix A; Positive HRESIMS: *m/z* 355.0789 [M + Na]^+^ (calcd for C_17_H_16_O_7_Na, 355.0788, Appendix A). 

### 4.5. Marfey’s Analysis of Compound **2**

Compound **2** (2.0 mg) was dissolved in 6 *N* HCl (2.0 mL) and heated at 100 °C for 24 h. The solutions were then evaporated to dryness and placed in a 4 mL reaction vial and treated with a 10 mg/mL solution of FDAA (200 μL) in acetone, followed by 1 M NaHCO_3_ (40 μL). The reaction mixtures were heated at 45 °C for 90 min, and the reactions were quenched by the addition of HCl (1 *N*, 40 *µ*L). In a similar fashion, the standard l-proline and d-proline were derivatized separately. The derivatives of the acid hydrolysate and the standard amino acids were subjected to RP HPLC analysis (Kromasil C18 column; 5 μM, 4.6 × 250 mm; 1.0 mL/min; UV detection at 340 nm), with a linear gradient of acetonitrile (30–40%) in water (TFA, 0.01%) over 30 min. The retention times for the authentic standards were as follows: l-proline derivative (8.91 min) and d-proline derivative (9.88 min). The absolute configuration of the chiral amino acid in **2** was determined by comparing the retention times.

### 4.6. Computational Details for NMR and ECD

The GMMX software tool was used to undertake the systematic conformational evaluations for **4** and **17,** utilizing the MMFF94 molecular mechanics force field. Gaussian 16 software was used to further improve the MMFF94 conformers, utilizing the M062X/6-31G(d) basis set level in gas for NMR calculations and B3LYP/6-31+G(d,p) basis set level in methanol, with a PCM model for ECD calculations. The shielding constants were calculated using the GIAO technique in chloroform, using the SMD solvent model and Gaussian function at mPW1PW91/6-31+G(d,p). A previously documented approach was used to calculate the ^1^H and ^13^C chemical shifts for the DP4+ probability analysis [77]. ECD spectra were stimulated in methanol with a Gaussian function at the B3LYP/6-311+G(2d,p) level using the PCM model, and 60 NStates were calculated. Boltzmann statistics were used to compute the equilibrium populations of the conformers at 298.15 K, based on their respective free energies (***Δ***G). The Boltzmann weighting of the key conformers was then used to construct the overall ECD spectra. UV correlation was used to correct the systematic mistakes in predicting the wavelength and excited-state energy [78].

### 4.7. Antimicrobial Assay

An antimicrobial assay was performed according to the Antimicrobial Susceptibility Testing Standards, outlined by the Clinical and Laboratory Standards Institute against MRSA (clinical strain from Chaoyang Hospital, Beijing, China), *Pseudomonas aeruginosa* (ATCC 15692), *Escherichia coli* (O57:H7), *Mycobacterium bovis* (ATCC35743), vancomycin-resistant *Enterococci faecalis* (VRE) (clinical strain from 309 Hospital, Beijing, China), and pathogen fungi *Candida albicans* SC5314. The protocol was performed as previously reported [58,59]. The positive controls were vancomycin against MRSA, *E. faecalis*, ciprofloxacin against *P. aeruginosa* and *E. coli*, amphotericin B for *C. albicans*, and rifampicin for *M. bovis*. All the experiments were performed in triplicate.

## 5. Conclusions

In summary, we isolated forty-five compounds from *A. fumigatus* H22, including six new compounds **1–4**, **17**, and **37**. The stereochemistry of the new compounds was determined by quantum calculations of NMR, ECD calculations and chemical derivatizations. Bioactivity screening indicated that compounds **5**, **8**, **10**, **11**, **16**, **21**, **23**, **29**–**38**, and **41** exhibited antimicrobial activities against MRSA, with MIC values ranging from 1.25 to 25 μM. Compound **8** also exhibited strong activity against *M. bovis,* with a MIC of 25 μM. To the best of our knowledge, this is the first report for the antimicrobial activities of compounds **5**, **10**, **11**, **16**, **30**, **31**, and **37**. The strains of *A. fumigatus* from ocean environments are a good source of antibacterial natural products, deserving further exploitation.

## Figures and Tables

**Figure 1 marinedrugs-20-00302-f001:**
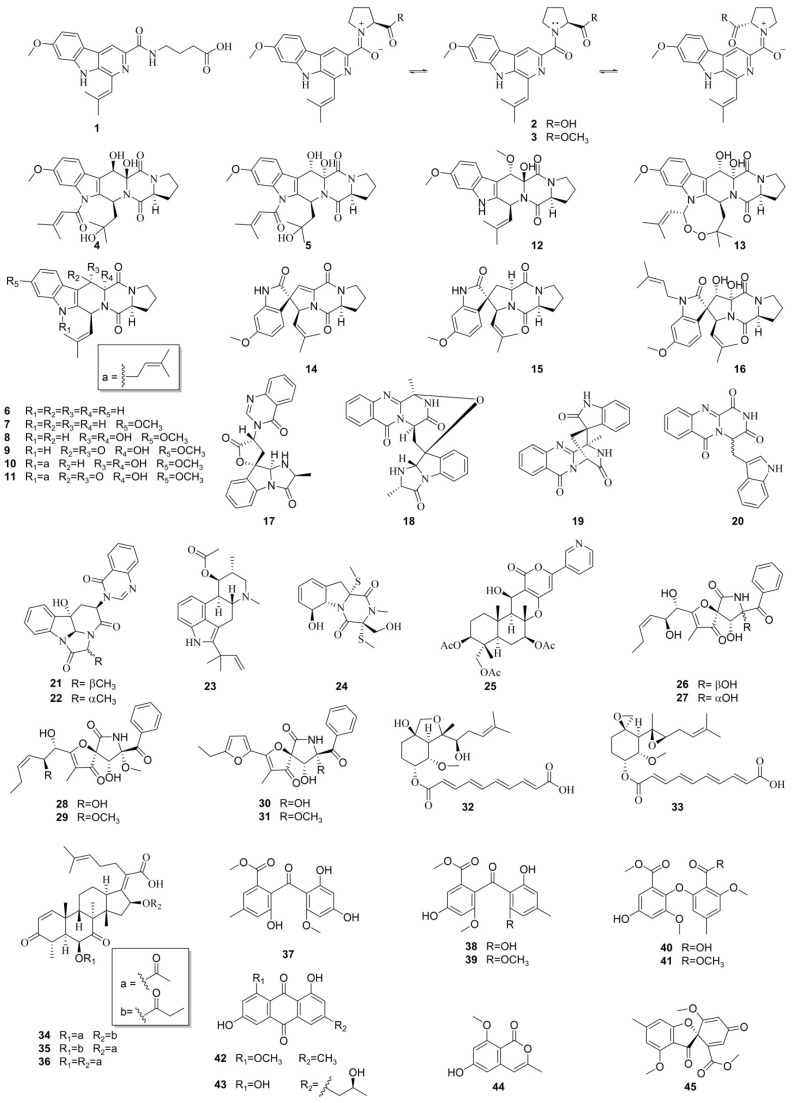
Structures of compounds **1**–**45**.

**Figure 2 marinedrugs-20-00302-f002:**
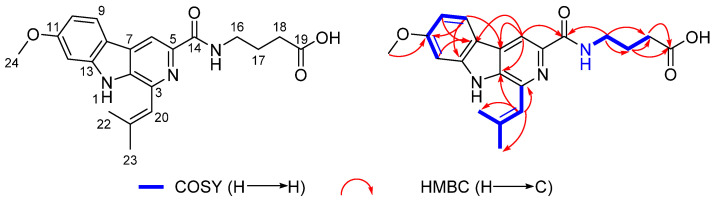
Key ^1^H–^1^H COSY and HMBC correlations of **1**.

**Figure 3 marinedrugs-20-00302-f003:**
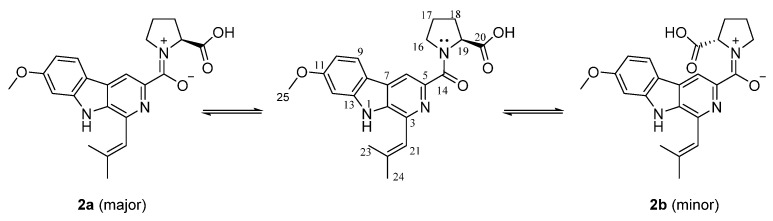
Scheme of the resonance structure of **2** and the chemical equilibrium between **2a** and **2b**.

**Figure 4 marinedrugs-20-00302-f004:**
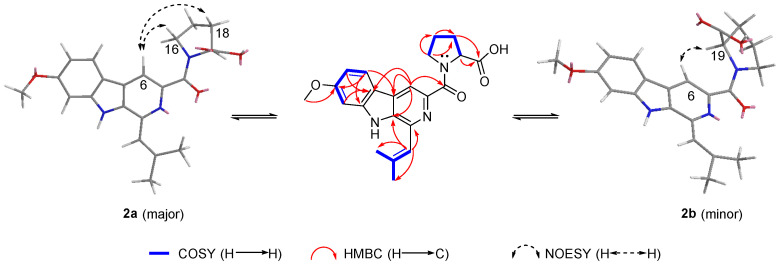
Key ^1^H-^1^H COSY, HMBC, and ROESY correlations of **2a** and **2b**.

**Figure 5 marinedrugs-20-00302-f005:**
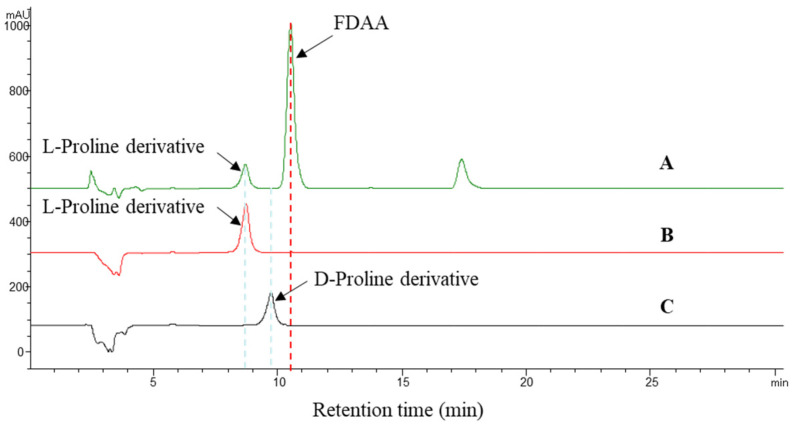
Advanced Marfey’s analysis of compound **2**. (**A**): The FDAA derivatives of the hydrolysate of **2**. (**B**,**C**): The retention times for the FDAA derivatives of l-proline and d-proline. The derivatives of the acid hydrolysate and the standard amino acids were subjected to RP HPLC analysis (Kromasil C18 column; 5 μM, 4.6 × 250 mm; 1.0 mL/min; UV detection at 340 nm) with a linear gradient of acetonitrile (30–40%) in water (TFA, 0.01%) over 30 min.

**Figure 6 marinedrugs-20-00302-f006:**
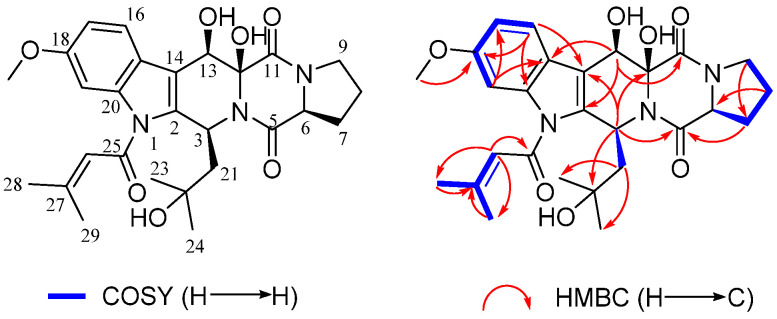
Key ^1^H-^1^H COSY, and HMBC correlations of **4**.

**Figure 7 marinedrugs-20-00302-f007:**
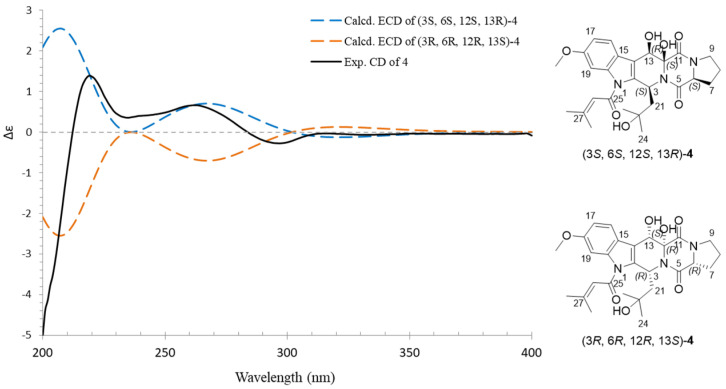
Experimental ECD spectra of compound **4** and the calculated ECD spectra of (3*S,* 6*S,* 12*S,* 13*R*)-**4** and (3*R,* 6*R,* 12*R,* 13*S*)-**4**.

**Figure 8 marinedrugs-20-00302-f008:**
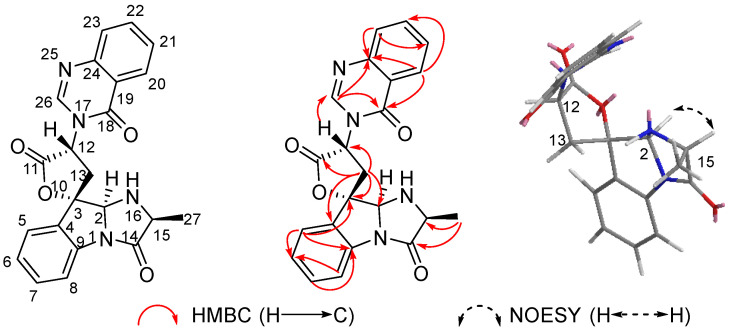
Key HMBC and NOESY correlations of compound **17**.

**Figure 9 marinedrugs-20-00302-f009:**
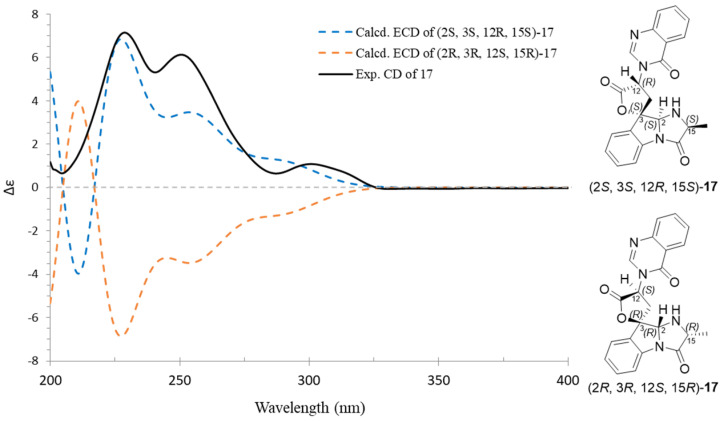
Experimental ECD spectra of compound **17** and the calculated ECD spectra of (2*S**,* 3*S**,* 12*R**,* 15*S*)-**17** and (2*R**,* 3*R**,* 12*S**,* 15*R*)-**17**.

**Figure 10 marinedrugs-20-00302-f010:**
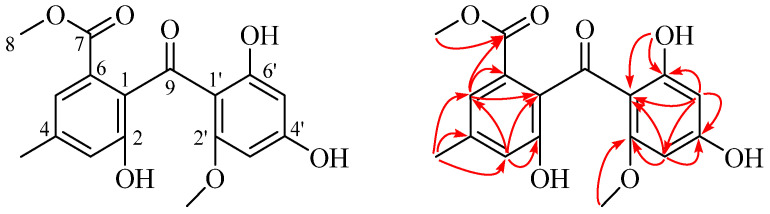
Key HMBC correlations of compound **37**.

**Table 1 marinedrugs-20-00302-t001:** ^1^H (500 MHz) and ^13^C NMR (125 MHz) data of compounds **1**–**3** in DMSO-*d*_6_.

Positions	1		2a		2b		3a		3b	
	*δ* _C_	*δ*_H_, Mult., (*J* in Hz)	*δ* _C_	*δ*_H_, Mult., (*J* in Hz)	*δ* _C_	*δ*_H_, Mult., (*J* in Hz)	*δ* _C_	*δ*_H_, Mult., (*J* in Hz)	*δ* _C_	*δ*_H_, Mult., (*J* in Hz)
1		11.85 br. s		11.83 br. s		11.64 br. s		11.92 br. s		11.60 br. s
2	135.1		134.4		134.2		134.3		134.3	
3	142.4		142.5		141.9		143.4		141.4	
5	158.3		158.2		159.7		158.4		158.1	
6	111.1	8.59 s	113.4	8.41 s	113.3	8.37 s	113.6	8.47 s	113.5	8.45 s
7	128.9		129.0		128.7		129.1		128.7	
8	115.0		114.8		114.9		114.8		114.9	
9	123.1	8.21 d (8.7)	123.2	8.22 d (8.7)	123.1	8.18 d (8.7)	123.4	8.25 d (8.7)	123.1	8.19 d (8.7)
10	110.2	6.91 dd (8.7, 2.3)	110.2	6.91 dd (8.7, 2.2)	110.0	6.89 dd (8.7, 2.2)	110.5	6.93 dd (8.7, 2.2)	110.0	6.89 dd (8.7, 2.2)
11	160.7		160.6		160.8		161.0		160.7	
12	94.8	7.05 d (2.3)	94.7	7.05 d (2.2)	94.7	7.04 d (2.2)	94.7	7.06 d (2.2)	94.8	7.04 d (2.2)
13	142.8		142.9		142.8		143.2		142.8	
14	164.7		166.0		166.6		165.5		165.8	
15		8.42 br. s								
16	38.3	3.39 q (7.4)	49.6	3.96 m	47.5	3.67 m	49.6	3.98 m	47.9	3.70 m
				3.88 m				3.91 m		
17	24.9	1.80 p (7.4)	25.2	1.94 m	21.9	1.92 m	25.2	1.96 m	21.8	1.92 m
				1.90 m		1.82 m		1.93 m		1.83 m
18	31.2	2.30 t (7.4)	28.6	2.26 m	31.2	2.29 m	28.6	2.28 m	31.3	2.29 m
				1.89 m		2.02 m		1.91 m		1.98 m
19	174.2		59.8	4.48 dd (8.8, 4.4)	60.4	5.30 dd (8.5, 3.6)	59.7	4.57 dd (8.6, 4.0)	60.8	5.18 dd (8.6, 4.5)
20	118.5	6.79 s	173.5		173.8		172.5		172.9	
21	138.7		118.5	6.76 s	119.0	6.66 s	118.0	6.75 s	119.5	6.59 s
22	27.3	2.08 s	138.1		138.0		138.1		138	
23	20.6	2.17 s	27.1	2.07 s	26.9	2.04 s	27.0	2.07 s	26.5	2.04 s
24	55.4	3.89 s	20.4	2.13 s	20.2	2.01 s	20.4	2.10 s	20.1	1.90 s
25			55.4	3.89 s	55.4	3.88 s	55.5	3.89 s	55.4	3.88 s
26							51.8	3.68 s	51.6	3.45 s

“m” means multiplet or overlapped with other signals.

**Table 2 marinedrugs-20-00302-t002:** ^1^H (500 MHz) and ^13^C NMR (125 MHz) data for **4** in chloroform-*d*.

Positions	*δ* _C_	*δ*_H_, Mult., (*J* in Hz)	Positions	*δ* _C_	*δ*_H_, Mult., (*J* in Hz)
2	137.0		17	111.2	6.90 dd (8.6, 2.2)
3	43.3	6.37 dd (9.5, 1.2)	18	157.7	
5	164.7		19	100.8	7.27 d (2.2)
6	59.8	4.32 dd (10.8, 6.0)	20	136.1	
7	29.6	2.51 m	21	39.5	2.29 dd (14.0, 9.5)
		1.95 m			2.14 dd (14.0, 1.2)
8	22.0	2.08 m	22	74.6	
		1.98 m	23	29.3	1.25 s
9	45.7	3.76 m	24	32.2	1.17 s
		3.65 m	25	165.5	
11	165.9		26	119.8	6.40 br. s
12	86.2		27	158.2	
13	68.4	5.13 s	28	27.4	2.11 s
14	114.3		29	21.2	2.21 s
15	122.3		18-OCH_3_	55.9	3.85 s
16	119.4	7.45 d (8.6)			

**Table 3 marinedrugs-20-00302-t003:** ^1^H (500 MHz) and ^13^C NMR (125 MHz) data of compound **17** in chloroform-*d*.

Positions	*δ* _C_	*δ*_H_, Mult., (*J* in Hz)	Positions	*δ*_C_, Type	*δ*_H_, Mult., (*J* in Hz)
2	82.6	5.82 s	14	172.4	
3	91.3		15	58.5	4.26 q (6.5)
4	134.4		18	161.0	
5	124.4	7.31 d (7.5)	19	121.9	
6	125.8	7.22 dd (8.0, 7.5)	20	126.8	8.29 d (8.1)
7	131.3	7.43 dd (8.0, 7.5)	21	128.2	7.58 dd (8.1, 7.5)
8	115.8	7.65 d (8.0)	22	135.4	7.85 dd (8.1, 7.5)
9	138.9		23	128.0	7.78 d (8.1)
11	170.9		24	148.1	
12	58.3	5.03 dd (10.5, 4.3)	26	145.6	8.11 s
13	33.4	3.68 dd (13.4, 4.3)	27	17.9	1.28 d (6.5)
		2.61 dd (13.4, 10.5)			

**Table 4 marinedrugs-20-00302-t004:** ^1^H (500 MHz) and ^13^C NMR (125 MHz) data of compound **37** in DMSO-*d*_6_.

Positions	*δ* _C_	*δ*_H_, Mult., (*J* in Hz)	Positions	*δ* _C_	*δ*_H_, Mult., (*J* in Hz)
1	127.2		1′	105.9	
2	153.1		2′	163.0	
3	120.1	6.88 s	3′	91.4	5.80 d (2.2)
4	138.1		4′	165.2	
5	120.2	7.19 s	5′	95.5	5.90 d (2.2)
6	130.3		6′	166.0	
7	166.0		4-CH_3_	20.8	2.30 s
8	51.9	3.64 s	2′-OCH_3_	55.7	3.26 s
9	198.1		6′-OH		13.55 s

**Table 5 marinedrugs-20-00302-t005:** Antibacterial assay results of monomeric compounds.

MIC (μM)			
Compound	MRSA ^a^	Compound	MRSA ^a^
**5**	5.00	**31**	5.00
**8**	2.50	**32**	25.0
**10**	20.0	**33**	2.50
**11**	1.25	**34**	1.25
**16**	10.00	**35**	1.25
**21**	25.00	**36**	1.25
**23**	12.50	**37**	1.25
**29**	10.00	**38**	1.25
**30**	5.00	**41**	5.00
Positive control	Vancomycin (1.00)		

^a^ MRSA: methicillin-resistant *Staphylococcus aureus*.

## Data Availability

Not applicable.

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
