# Peer review of "Discovery of Anti-MRSA Secondary Metabolites from a Marine-Derived Fungus Aspergillus fumigatus"

_marinedrugs, 2022, doi:10.3390/md20050302_

Round 1
Reviewer 1 Report
This manuscript reported chemical investigation of marine derived fungus Aspergillus fumigatus H-22 collected at middle seawater from Western Pacific, which led to isolation of 45 metabolites, where six compounds were first time discovered in this study. Antibacterial evaluation against MRSA has carried out on these isolated metabolites.
Structure discussion for compound 4 should state the difference between 4 and 5. Also, Mosher's method can be used for 4 to determine 13-OH configuration.
Marfey's method required citation.
Author Response
We have thoroughly revised our manuscript as required. The followings are detailed corrections and responses (highlighted in yellow in the revised manuscript)

Reviewer 2 Report
This manuscript reports 6 novel and 39 known compounds that were isolated from a marine-derived fungus. Those compounds were then examined in several bioactivity tests, leading to significant antimicrobial activities against MRSA. The methods and analytical strategy of isolation and structure elucidation are appropriately described. The compounds are expected to be good seeds for MRSA problems. Therefore, the reviewer would recommend publication in the journal, if the authors update the following minor points.
(1) In Figure 1 of page 3, labels of atoms are not consistent in fonts. Please take care of them.
(2) In line 74 of page 4, there is a misspelling: "exchangebale" should be "exchangeable".
(3) In line 78 of page 4, the chemical shift 158.0 is not seen in Table 1. Please check it again.
(4) In line 125 of page 6, "is existed" sounds strange to the reviewer. An intransitive verb is not used in the passive voice.
(5) In line 151 of page 7, is "possing" correct? It should be "possessing" perhaps.
(6) In lines 159-164 of page 7, chemical shifts used in discussion are not exactly the same with those in Table 4.
(7) In line 185 of page 7, the authors say "the experimental ECD was match better with that of (3S, 6S, 12S, and 13R)-4" (please also notice English is wrong), but it's not obvious to the reviewer, because the fitting situations differ between 200-250 and 250-300 nm. The authors should explain how we can analyze the data to lead to the conclusion. It'd be better if discussion gets related to UV spectrum, because ECD has a close correlation with UV. The reviewer also wonder how the functional PBE1BPE was chosen. How were the results from other functionals: did different functionals give the same conclusion?
(8) In line 195 of page 8, hyphens for 1H-NMR and 13C-NMR are not necesssary. They have already appeared as 1H NMR and 13 NMR, respectively.
(9) In lines 195-203 of page 8, some 1H values are combined with their 13C, but others are not. Please be constant.
(10) In lines 220-227 of page 9, discussion on ECD in Figure should be correlated with UV data.
(11) In line 236 of page 10, "moietys" should be "moieties".
(12) In line 240 of page 10, discussion about sulochrin (38) needs the reference.
(13) In line 245 of page 10, C-2 is missing in Figure 10. Is this about C-6?
(14) In Table 4 of page 10, some numbers require proper significant figures. Concretely, 5.8 and 5.9 should be 5.80 and 5.90, respectively.
(15) In line 274 of page 11, "compounds 8" should be "compound 8".
Author Response

(The authors gave the same response as above.)

Reviewer 3 Report
This manuscript discusses well on the structures of six new molecules and anti-MDR activities of isolated molecules. The reviewer found only minor points.
On the stereochemical difference between 2a and 2b, the authors observed strong NOEs as in the lines 130-132. The reviewer confirmed the cross peaks, but they are just normal. In addition, the reviewer also wonders if it is possible to clearly separate the NOEs between two forms? One more thing is the nomenclature of these forms. Should we call them as rotamers or isomers?
In the antimicrobial assay, the authors tested only with drug resistant strains, however, it is better to test also against drug sensitive strains to see the specific effects against drug resistant strains.
Methylation (using TMS diazomethane) of compound 2 results in the formation of 3 for the proof of stereochemical assignment.
In the name of compound 17, epi of “2-epi-…“ may be italic.
There are mistakes such as: Page 10, lines 232 & 250 in the titles of Tables 3 and 4: from “Compounds” to “compound”. Another example is line 464. It is recommended that the authors use commercial service to polish the text.
Author Response

(The authors gave the same response as above.)
